# Clec11a/osteolectin is an osteogenic growth factor that promotes the maintenance of the adult skeleton

**Rui Yue[1], Bo Shen[1], Sean J Morrison[1,2]***

[1]Department of Pediatrics and Children's Research Institute, University of Texas Southwestern Medical Center, Dallas, United States; [2]Howard Hughes Medical Institute, University of Texas Southwestern Medical Center, Dallas, United States

**Abstract** Bone marrow stromal cells maintain the adult skeleton by forming osteoblasts throughout life that regenerate bone and repair fractures. We discovered that subsets of these stromal cells, osteoblasts, osteocytes, and hypertrophic chondrocytes secrete a C-type lectin domain protein, Clec11a, which promotes osteogenesis. *Clec11a*-deficient mice appeared developmentally normal and had normal hematopoiesis but reduced limb and vertebral bone. *Clec11a*-deficient mice exhibited accelerated bone loss during aging, reduced bone strength, and delayed fracture healing. Bone marrow stromal cells from *Clec11a*-deficient mice showed impaired osteogenic differentiation, but normal adipogenic and chondrogenic differentiation. Recombinant Clec11a promoted osteogenesis by stromal cells in culture and increased bone mass in osteoporotic mice in vivo. Recombinant human Clec11a promoted osteogenesis by human bone marrow stromal cells in culture and in vivo. Clec11a thus maintains the adult skeleton by promoting the differentiation of mesenchymal progenitors into mature osteoblasts. In light of this, we propose to call this factor Osteolectin.

***For correspondence:** sean.morrison@utsouthwestern.edu

## Introduction

Fate mapping studies in vivo show there are multiple distinct waves of mesenchymal progenitors that form skeletal tissues during development and then maintain the skeleton throughout adulthood (*Liu et al., 2013*; *Maes et al., 2010*; *Mizoguchi et al., 2014*; *Park et al., 2012*; *Takashima et al., 2007*; *Worthley et al., 2015*; *Zhou et al., 2014a*). These include Osterix[+] cells that give rise to osteoblasts, osteocytes, and stromal cells in developing bones (*Liu et al., 2013*; *Maes et al., 2010*; *Mizoguchi et al., 2014*), Nestin-CreER-expressing cells that transiently form osteoblasts and bone marrow stromal cells in the early postnatal period (*Méndez-Ferrer et al., 2010*; *Ono et al., 2014a*; *Takashima et al., 2007*), *Grem1*-expressing cells that form osteoblasts, chondrocytes, and stromal cells postnatally (*Worthley et al., 2015*) and Leptin Receptor (LepR)-expressing stromal cells that are the major source of bone and adipocytes in adult mouse bone marrow (*Mizoguchi et al., 2014*; *Zhou et al., 2014a*). Osterix[+] osteogenic progenitors also persist periosteally, on the outer surface of adult bones, where they help to repair bone injuries (*Maes et al., 2010*).

Bone marrow stromal cells include skeletal stem cells (SSCs) as well as multiple other populations of mesenchymal progenitors (*Chan et al., 2015*). SSCs are multipotent progenitors that form fibroblast colonies in culture (CFU-F) with the potential to differentiate into osteoblasts, chondrocytes, and adipocytes (*Bianco and Robey, 2015*; *Friedenstein et al., 1970*). Bone marrow CFU-F can be identified based on the expression of CD146, CD271, VCAM-1, and Thy-1 in humans, or LepR, PDGFRα, CD51, and/or CD105 in mice, as well as the lack of expression of hematopoietic and endothelial markers (*Chan et al., 2009*, *2015*; *James et al., 2015*; *Mabuchi et al., 2013*;

*Morikawa et al., 2009*; *Omatsu et al., 2010*; *Park et al., 2012*; *Sacchetti et al., 2007*; *Zhou et al., 2014a*). CFU-F are enriched among bone marrow stromal cells that express high levels of the hematopoietic growth factors *Scf* (*Zhou et al., 2014a*) and *Cxcl12* (*Ding and Morrison, 2013*; *Omatsu et al., 2014*; *Sugiyama et al., 2006*).

Multiple growth factor families promote osteogenesis including Wnts (*Cui et al., 2011*; *Krishnan et al., 2006*), Bone Morphogenetic Proteins (BMPs) (*Nakamura et al., 2007*; *Rahman et al., 2015*), and Insulin-like Growth Factors (*Yakar and Rosen, 2003*). However, these factors have broad effects on many tissues, precluding their systemic administration to promote osteogenesis. Sclerostin, a Wnt signaling inhibitor that is locally produced by osteocytes, negatively regulates bone formation (*Li et al., 2005*). Sclerostin inhibitors can be administered systemically to promote bone formation (*McClung et al., 2014*). Factors secreted by bone marrow stromal cells promote osteogenesis (*Chan et al., 2015*), though the full repertoire of such factors remains to be identified.

Osteoporosis is a progressive bone disease characterized by decreased bone mass and increased fracture risk (*Harada and Rodan, 2003*). Aging, estrogen insufficiency, long-term glucocorticoid use, and mechanical unloading all contribute to the development of osteoporosis (*Harada and Rodan, 2003*). Most existing osteoporosis therapies involve antiresorptive agents, such as bisphosphonates (*Black et al., 1996*; *Liberman et al., 1995*) and estrogens (*Michaelsson et al., 1998*), which reduce the rate of bone loss but do not promote new bone formation. Teriparatide, a small peptide derived from human parathyroid hormone (PTH; amino acids 1–34) is used clinically to promote the formation of new bone (*Neer et al., 2001*). Nonetheless, some patients cannot take Teriparatide (*Kraenzlin and Meier, 2011*) and its use is limited to two years because of a potential risk of osteosarcoma (*Neer et al., 2001*).

Clec11a (C-type lectin domain family 11, member A) is a secreted sulfated glycoprotein that is expressed in the bone marrow and can promote colony formation by human hematopoietic progenitors in culture (*Bannwarth et al., 1998*, *1999*; *Hiraoka et al., 1997*; *Mio et al., 1998*). The plasma level of human Clec11a correlates with hemoglobin level (*Keller et al., 2009*; *Ouma et al., 2010*) and increases in patients after bone marrow transplantation (*Ito et al., 2003*). As a result, Clec11a has been considered a hematopoietic growth factor. However, *Clec11* is also expressed in skeletal tissues (*Hiraoka et al., 2001*) and the physiological function of Clec11a in vivo has not yet been tested.

## Results

### *Clec11a* is expressed by subsets of bone marrow stromal cells, osteoblasts, osteocytes and hypertrophic chondrocytes

Reanalysis of our published microarray data (*Ding et al., 2012*) revealed that among enzymatically dissociated bone marrow cells, *Clec11a* was significantly more highly expressed by *Scf*-GFP[+]-CD45[-]Ter119[-]CD31[-] stromal cells (more than 90% of which are also LepR[+] [*Zhou et al., 2014a*]) and Col2.3-GFP[+]CD45[-]Ter119[-]CD31[-] osteoblasts as compared to VE-cadherin[+] endothelial cells and unfractionated cells (*Figure 1A*). By RNA sequencing, *Clec11a* transcripts were at least 100-fold more abundant in PDGFRα[+]CD45[-]Ter119[-]CD31[-] bone marrow stromal cells (more than 90% of which are LepR[+] [*Zhou et al., 2014a*]) as compared to unfractionated bone marrow cells (*Figure 1B*). A systematic analysis of *Clec11a* expression in bone marrow cells by quantitative reverse transcription PCR (qRT-PCR) showed that *Clec11a* was highly expressed by LepR[+]CD45[-]Ter119[-]CD31[-] stromal cells and Col2.3-GFP[+]CD45[-]Ter119[-]CD31[-] osteoblasts but not by hematopoietic cells (*Figure 1C*). *Clec11a* was also expressed at a very low level by B cell progenitors in the bone marrow and T cells in the spleen (*Figure 1C*).

Published gene expression profile data were consistent with our results. RNA-seq analysis showed 80-fold higher levels of *Clec11a* in bone fragments that contain osteoblasts and osteocytes as compared to whole bone marrow or skeletal muscle cells (*Ayturk et al., 2013*). *Clec11a* expression is 9-fold higher in Grem1[+] SSCs as compared to Grem1[-] stromal cells (see GSE57729 from [*Worthley et al., 2015*]). LepR[+] stromal cells are enriched for *Grem1* expression (see GSE33158 from [*Ding et al., 2012*]).

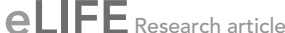

**Figure 1.** *Clec11a* deficient mice were grossly developmentally normal and had normal hematopoiesis. (**A–C**) *Clec11a* expression analysis by microarray, RNA-seq and qRT-PCR. Whole bone marrow cells, VE-Cadherin+ bone marrow endothelial cells, bone marrow stromal cells (*Scf*-GFP+CD45-Ter119-CD31- for microarray, PDGFRα+CD45-Ter119-CD31- for RNA-seq, and LepR+CD45-Ter119-CD31- for qPCR), *Col2.3*-GFP+CD45-Ter119-CD31- osteoblasts and hematopoietic cell populations were sorted from enzymatically dissociated femur bone marrow of two

*Figure 1 continued on next page*

*Figure 1 continued*

month-old mice, followed by microarray (A), RNA-seq (B) and qPCR (C) analysis. The statistical significance of differences was assessed using one-way ANOVAs with Dunnett's multiple comparison tests (n = 2–3 mice/genotype from at least two independent experiments). (D) The position of images from femur sections. (E–H) Confocal analysis of isotype control or anti-Clec11a antibody staining in the femur metaphysis (E and F) and diaphysis (G and H) of *Prrx1-Cre; tdTomato* mice. Bone was imaged by second harmonic generation (SHG). Bone marrow stromal cells (Fi), osteoblasts (Fii), hypertrophic chondrocytes (Fiii) and osteocytes (H) are marked by arrows (n = 3 mice per genotype, total, from three independent experiments). (I and J) Confocal analysis of isotype control (I) or anti-Clec11a antibody staining (J) in the femur metaphysis of *Lepr-Cre; tdTomato* mice. LepR$^+$ bone marrow stromal cells are marked by arrows (n = 3 mice per genotype, total, from three independent experiments). (K) Representative images of two month-old control and *Clec11a$^{-/-}$* mice. (L) Body mass of 2 and 10 month-old mice (n = 3–8 mice per genotype, total, from at least three independent experiments for all data in panels L–T). (M) Cellularity of the bone marrow and spleen. (N–T) Flow cytometric analysis of the frequencies of myeloid cells (N), erythroid progenitors (O), T cells (P), B cells (Q), hematopoietic stem cells (R), multipotent progenitors (S) and restricted progenitors (T) in the bone marrow and spleen of *Clec11a$^{-/-}$* mice and sex-matched littermate controls. The statistical significance of differences among genotypes was assessed using two-tailed Student's t tests. Data represent mean ± SD: *p<0.05, **p<0.01, ***p<0.001. The source data are in *Figure 1—source data 1*.

The following source data and figure supplement are available for figure 1:

**Source data 1.** Data for *Figure 1A* and *Figure 1—figure supplement 1*.
**Figure supplement 1.** Generation of *Clec11a$^{-/-}$* mice and hematopoietic analysis.

To assess the Clec11a protein expression, we stained femur sections from eight week-old *Prrx1-Cre; tdTomato* reporter mice with a commercial polyclonal antibody against Clec11a. *Prrx1-Cre* recombines in limb bone marrow mesenchymal cells including SSCs, other bone marrow stromal cells, osteoblasts, osteocytes, and chondrocytes (*Ding and Morrison, 2013*; *Greenbaum et al., 2013*; *Logan et al., 2002*; *Yue et al., 2016*). In two month-old mice, most of the Clec11a staining was observed in and around the trabecular bone in the femur metaphysis (*Figure 1F*) as well as in cortical bone of the proximal femur (*Figure 1H*). This staining pattern was specific for Clec11a as we did not observe any staining in bone marrow sections from *Clec11a* deficient mice (*Figure 1—figure supplement 1A and B*; see below) or in sections stained with an isotype control in place of the anti-Clec11a antibody (*Figure 1E,G, and I*).

Within the distal femur metaphysis of *Prrx1-Cre; tdTomato* mice we observed Clec11a staining adjacent to Tomato$^+$ stromal cells in the bone marrow near trabecular bone (*Figure 1Fi*), Tomato$^+$ osteoblasts lining trabecular bone surfaces (*Figure 1Fii*), and Aggrecan$^+$ hypertrophic chondrocytes (*Figure 1Fiii*; *Figure 1—figure supplement 1D*). In the cortical bone matrix, we also observed Clec11a staining amongst osteocytes (*Figure 1H*). We did not detect Clec11a expression by bone marrow stromal cells, osteoblasts, or osteocytes in much of the diaphysis. Nonetheless, given the tendency of secreted growth factors to concentrate in regions of extracellular matrix, especially within the bone matrix (*Hauschka et al., 1986, 1988*; *Mohan and Baylink, 1991*), Clec11a may be more broadly expressed by bone marrow stromal cells than is evident from the antibody staining pattern.

The position and morphology of the metaphyseal bone marrow stromal cells that were associated with Clec11a staining (*Figure 1Fi*) suggested that these cells included LepR$^+$ cells. To test this, we stained femur sections from eight week-old *Lepr-Cre; tdTomato* mice with anti-Clec11a antibody. We observed Clec11a staining adjacent to a subset of Tomato$^+$ stromal cells near trabecular bone in the metaphysis (*Figure 1J*) but not by Tomato$^+$ stromal cells throughout most of the diaphysis. The Clec11a staining in LepR$^+$ cells in the metaphysis (*Figure 1J*) was clearly above background (*Figure 1I*) but was dimmer than observed in and around bone matrix (*Figure 1F and H*). It is unclear whether this reflects the lower Clec11a expression by the LepR$^+$ cells or whether Clec11a is bound and concentrated by bone matrix.

We observed a similar Clec11a expression pattern in vertebrae, with anti-Clec11a antibody staining in and around the vertebral trabecular bone near the growth plate as well as in cortical bone (*Figure 1—figure supplement 1C*). Our data thus indicate that Clec11a is expressed by subsets of LepR$^+$ bone marrow stromal cells, osteoblasts, and hypertrophic chondrocytes in the metaphysis as well as by osteocytes in certain regions of cortical bone.

## *Clecl11a* is not required for hematopoiesis in normal mice

To test the physiological function of *Clec11a* we used CRISPR-Cas9 to generate a *Clec11a* mutant allele (*Clec11a$^{-/-}$*) by deleting the second exon of *Clec11a* (*Figure 1—figure supplement 1E*). This was predicted to be a strong loss of function as exon 2 deletion introduced a frame shift that created a premature stop codon in exon 3 (*Figure 1—figure supplement 1F*). The predicted mutant protein did not contain any of the domains that are thought to be functionally important in Clec11a, including the polyglutamic acid sequence, the alpha-helical leucine zipper, or the C-type lectin domain (*Figure 1—figure supplement 1F*). Germline transmission of the mutant allele was confirmed by PCR and sequencing of genomic DNA (*Figure 1—figure supplement 1G*). *Clec11a* deficiency was also confirmed by qPCR analysis of bone marrow LepR$^+$CD45$^-$Ter119$^-$CD31$^-$ cells from *Clec11a$^{-/-}$* mice (*Figure 1—figure supplement 1H*) and by *Clec11a* cDNA sequencing to confirm exon two deletion (*Figure 1—figure supplement 1I*), and by the loss of Clec11a from the plasma *Clec11a$^{-/-}$* mice (*Figure 1—figure supplement 1J*). Immunofluorescence analysis of femur sections with an anti-Clec11a polyclonal antibody suggested a complete loss of Clec11a protein from *Clec11a$^{-/-}$* mice (*Figure 1—figure supplement 1A and B*).

*Clec11a$^{-/-}$* mice were born with Mendelian frequency (*Figure 1—figure supplement 1K*) and appeared grossly normal (*Figure 1K*), with normal body mass at 2 and 10 months of age (*Figure 1L*). White blood cell, red blood cell, and platelet counts were normal in 2, 10, and 16 month-old *Clec11a$^{-/-}$* mice (*Figure 1—figure supplement 1L–N*). Two and 10-month old *Clec11a$^{-/-}$* mice also had normal bone marrow and spleen cellularity (*Figure 1M*), as well as normal frequencies of Mac1$^+$Gr1$^+$ myeloid cells, Ter119$^+$CD71$^+$ erythroid progenitors, CD3$^+$ T cells, and B220$^+$ B cells in the bone marrow and spleen (*Figure 1N–Q*). *Clec11a$^{-/-}$* mice had normal frequencies of CD150$^+$-CD48$^-$Lineage$^-$Sca-1$^+$c-kit$^+$ HSCs (*Kiel et al., 2005*) and CD150$^-$CD48$^-$Lin$^-$Sca-1$^+$c-kit$^+$multipotent progenitors (MPPs) (*Kiel et al., 2008*; *Oguro et al., 2013*) in the bone marrow and spleen (*Figure 1R and S*), as well as normal frequencies of CD34$^+$FcγR$^+$Lin$^-$Sca-1$^-$c-kit$^+$granulocyte-macrophage progenitors (GMPs), CD34$^-$FcγR$^-$Lin$^-$Sca-1$^-$c-kit$^+$megakaryocyte-erythrocyte progenitors (MEPs), CD34$^+$FcγR$^-$Lin$^-$Sca-1$^-$c-kit$^+$ common myeloid progenitors (CMPs) (*Akashi et al., 2000*) and Flt3$^+$IL7Rα$^+$Lin$^-$Sca-1$^{low}$c-kit$^{low}$common lymphoid progenitors (CLPs) (*Kondo et al., 1997*) in the bone marrow (*Figure 1T*). Bone marrow from two month-old *Clec11a$^{-/-}$* mice gave long-term multilineage reconstitution upon transplantation into irradiated mice with normal levels of donor cell reconstitution (*Figure 1—figure supplement 1O–R*). *Clec11a* is therefore not required for normal hematopoiesis in adult mice.

Human recombinant Clec11a increases erythroid (BFU-E) and myeloid (CFU-G/M/GM) colony formation by human bone marrow cells when added to culture along with EPO or GM-CSF, respectively (*Hiraoka et al., 1997, 2001*). In cultures of mouse bone marrow cells, recombinant mouse Clec11a did not significantly increase BFU-E colony formation when added along with EPO and only slightly increased CFU-G/M/GM colony formation when added along with GM-CSF (*Figure 1—figure supplement 1S and T*).

## *Clec11a* is required for normal levels of osteogenesis in vivo

To test whether Clec11a regulates osteogenesis we performed micro-CT analysis of the distal femur from sex-matched littermates. In no case did we observe any significant difference between *Clec11a$^{+/+}$* and *Clec11a$^{+/-}$* mice (data not shown), so samples from these mice were combined as controls. We always compared sex-matched littermates within individual experiments, using paired statistical tests to assess the significance of differences across multiple independent experiments. Trabecular bone volume was significantly reduced (by 24 ± 18%) in two month-old *Clec11a$^{-/-}$* mice as compared to littermate controls (*Figure 2A and D*). The *Clec11a$^{-/-}$* mice had significantly reduced trabecular bone thickness, increased trabecular spacing, and decreased connectively density and bone mineral density (*Figure 2D–I*). With the exception of the reduction in bone mineral density, these defects seemed to worsen with age as 10 and 16 month-old *Clec11a$^{-/-}$* mice exhibited more profound reductions in trabecular bone volume (62 ± 27% and 64 ± 11%, respectively), trabecular number, trabecular thickness and connectivity density, as well as increased trabecular spacing (*Figure 2B–H*).

MicroCT analysis of cortical bone parameters in the femur diaphysis from sex-matched littermates did not show significant differences between *Clec11a$^{-/-}$* and control mice at 2 or 10 months of age



**Figure 2.** Clec11a is necessary for osteogenesis in limb bones and vertebrae. (A–C) MicroCT images of trabecular bone in the distal femur metaphysis of two month-old (A), 10 month-old (B) and 16 month-old (C) *Clec11a*$^{-/-}$ mice and sex-matched littermate controls. (D–I) MicroCT analysis of trabecular bone volume/total volume (D), trabecular number (E), trabecular thickness (F), trabecular spacing (G), connectivity density (H) and bone mineral density (I)) in the distal femur metaphysis of 2, 10 and 16 month-old *Clec11a*$^{-/-}$ mice and sex-matched littermate controls (n = 4–9 mice/genotype from at least

*Figure 2 continued on next page*

*Figure 2 continued*

four independent experiments). (**J–L**) MicroCT images of trabecular bone from the ventral L3 lumbar vertebrae of two month-old (**J**), 10 month-old (**K**) and 16 month-old (**L**) *Clec11a⁻/⁻* mice and sex-matched littermate controls. (**M–R**) MicroCT analysis of trabecular bone parameters in the L3 vertebral bodies of 2, 10 and 16 month-old *Clec11a⁻/⁻* mice and sex-matched littermate controls (n = 3–5 mice per genotype, total, from at least three independent experiments). (**S–U**) Representative calcein double labeling images (**S**) with quantification of the trabecular bone mineral apposition (**T**) and trabecular bone formation (**U**) rates in the femur metaphysis of 2 and 10 month-old mice (n = 4 mice per genotype, total, from four independent experiments). (**V**) Bone resorption analysis by measuring the deoxypyridinoline/creatinine ratio in urine (n = 4 mice/genotype from four independent experiments). The statistical significance of differences among genotypes was assessed using two-tailed Student's paired t tests. Data represent mean ± SD (*p<0.05, **p<0.01, ***p<0.001). All means included data from male and female mice but within individual experiments sex-matched littermates were compared by paired tests. The source data are in *Figure 2—source data 1*.

The following source data and figure supplement are available for figure 2:

**Source data 1.** Data for *Figure 2* and *Figure 2—figure supplement 1*.
**Figure supplement 1.** Further skeleton analysis in *Clec11a⁻/⁻* mice.

(*Figure 2—figure supplement 1A and B*). However, 16 month-old *Clec11a⁻/⁻* mice exhibited significantly reduced cortical bone area, cortical area/total area ratio, and cortical thickness as compared to controls (*Figure 2—figure supplement 1C–H*). The femur length was slightly but significantly reduced in 2, 10 and 16 month-old *Clec11a⁻/⁻* mice as compared to littermate controls (*Figure 2—figure supplement 1I*). When we tested the mechanical strength of bones using a three point bending test we found significantly reduced peak load and fracture energy in the femur diaphysis of 2, 10, and 16 month-old *Clec11a⁻/⁻* as compared to sex-matched littermate control mice (*Figure 2—figure supplement 1J–L*).

Micro-CT analysis of L3 vertebrae as a whole, including both cortical and trabecular bone, showed a trend toward reduced bone volume in *Clec11a⁻/⁻* as compared to sex-matched littermate controls, though the difference was not statistically significant (*Figure 2—figure supplement 1M–R*). Trabecular bone volume was significantly reduced in the L3 vertebral body of *Clec11a⁻/⁻* as compared to controls at 2, 10, and 16 months of age (*Figure 2J–M*). We also observed significantly reduced trabecular number and significantly increased trabecular spacing in L3 vertebrae from 2, 10 and 16 month-old *Clec11a⁻/⁻* as compared to littermate controls (*Figure 2N–P*). Clec11a is therefore required to maintain limb and vertebral bone.

Alizarin red/alcian blue double staining at postnatal day 3 did not reveal any significant differences between *Clec11a⁻/⁻* and littermate control mice (*Figure 2—figure supplement 1S*). This suggests that Clec11a is not required for fetal skeletal development.

We performed calcein double labeling to assess the rate of trabecular bone formation (*Figure 2S*). The trabecular bone mineral apposition and trabecular bone formation rates were both significantly decreased in the femur metaphysis of 2 and 10 month-old *Clec11a⁻/⁻* as compared to sex-matched littermate control mice (*Figure 2T* and U; 16 month-old mice were not assessed in these experiments). In contrast, the urinary bone resorption marker deoxypyridinoline did not significantly differ between *Clec11a⁻/⁻* and littermate controls (*Figure 2V*). This suggests that the difference in trabecular bone volume between *Clec11a⁻/⁻* and littermate control mice reflected reduced bone formation, not a change in bone resorption.

## Clec11a is necessary for mesenchymal progenitor differentiation into mature osteoblasts

To assess the mechanism by which Clec11a promotes osteogenesis in vivo we used multiple approaches to test whether it regulates the maintenance, proliferation, or differentiation of mesenchymal progenitors in the bone marrow. The frequency of LepR⁺CD45⁻Ter119⁻Tie2⁻ stromal cells did not significantly differ between the bone marrow of *Clec11a⁻/⁻* and littermate control mice (*Figure 3—figure supplement 1B*). We also did not detect any difference in the rate of BrdU incorporation by these cells in vivo (*Figure 3—figure supplement 1H*) or the percentage of these cells that stained positively for activated caspase 3/7 (data not shown).

We also examined a series of mesenchymal stem and progenitor cell populations that had been identified in a prior study of postnatal day 3 bone marrow based on expression of CD51 and other markers (*Chan et al., 2015*). In 10 month-old bone marrow we were able to identify 5 of the 8 cell populations that had been identified in the neonatal bone marrow (*Figure 3—figure supplement 1A*). All of these cell populations were uniformly or nearly uniformly positive for LeprR expression as nearly all CD51[+] bone marrow stromal cells were LepR[+] and vice versa (*Figure 3—figure supplement 1A*). Consistent with the data on LepR[+] cells above, we did not detect any effect of *Clec11a* deficiency on the frequency of these cell populations (*Figure 3—figure supplement 1C–G*), their rate of BrdU incorporation (*Figure 3—figure supplement 1I–M*), or the percentage of cells that stained positively for activated caspase 3/7 (data not shown). Clec11a is therefore not required in vivo for the maintenance, survival, or the proliferation of bone marrow mesenchymal progenitors.

To functionally assess this conclusion, we cultured at clonal density enzymatically dissociated femur bone marrow cells from *Clec11a[-/-]* and sex-matched littermate control mice at 2 and 10 months of age. We observed no difference in the frequency of cells that formed CFU-F colonies or in the number of cells per colony (*Figure 2—figure supplement 1T and U*). Clec11a is therefore not required for the maintenance of CFU-F in vivo or for their proliferation in culture.

To test whether Clec11a is required for the differentiation of mesenchymal progenitors, we cultured CFU-F from *Clec11a[-/-]* and littermate control bone marrow at clonal density, then replated equal numbers of *Clec11a[-/-]* or control cells into osteogenic, adipogenic, or chondrogenic culture conditions. Consistent with the decreased osteogenesis in vivo, bone marrow stromal cells from *Clec11a[-/-]* mice gave rise to significantly fewer cells with alkaline phosphatase (a marker of mature osteoblasts and pre-adipocytes) or alizarin red (a marker of mineralization by mature osteoblasts) staining as compared to control cells under osteogenic culture conditions (*Figure 3A–D*). qRT-PCR analysis of cells in these cultures showed that *Clec11a* deficiency did not significantly affect the expression of *Sp7 (Osterix)*, *Runx2*, or *Col1a1* but did significantly reduce the expression of Integrin binding sialoprotein (*Ibsp*) and Dentin matrix protein 1 (*Dmp1*) (*Figure 3—figure supplement 1N*). *Sp7*, *Runx2*, and *Col1a1* are broadly expressed by immature osteogenic progenitors and osteoblasts (*Jikko et al., 1999*; *Kulterer et al., 2007*; *Nakashima et al., 2002*) whereas *Ibsp* and *Dmp1* mark mature osteoblasts (*Kalajzic et al., 2005*). Clec11a was thus required for the differentiation of bone marrow mesenchymal progenitors into mature osteoblasts.

Under adipogenic (*Figure 3E and F*) and chondrogenic (*Figure 3G and H*) culture conditions we did not detect any difference between *Clec11a[-/-]* and control cells in terms of Oil Red O or Toluidine blue staining. Consistent with this, the number of Perilipin[+] adipocytes (*Figure 3I–K*) and Safranin O[+] chondrocytes (*Figure 3L–N*) in femur sections from two month-old mice did not differ between *Clec11a[-/-]* and sex matched littermate control mice. However, we did observe significantly more adipocytes in the femur sections from 10 month-old *Clec11a[-/-]* mice as compared to littermate controls (*Figure 3—figure supplement 2A–C*). The number of chondrocytes did not significantly differ between *Clec11a[-/-]* and littermate controls at 10 months of age (*Figure 3—figure supplement 2D–F*).

To test whether Clec11a is necessary for the differentiation of mesenchymal progenitors into mature osteoblasts in vivo, we cultured CFU-F from adult bone marrow at clonal density, then seeded equal numbers *Clec11a[-/-]* or control cells into collagen sponges and transplanted them subcutaneously into immunocompromised NSG mice (*Figure 3—figure supplement 1O*). Eight weeks after transplantation, ossicles generated from *Clec11a[-/-]* cells contained significantly less bone as compared to the ossicles generated from control cells (*Figure 3—figure supplement 1P*). In contrast, the number of adipocytes per unit area did not differ between ossicles containing *Clec11a[-/-]* as compared to control cells (*Figure 3—figure supplement 1Q*).

## Clec11a is necessary for normal fracture healing

We performed mid-diaphyseal femur fractures in two month-old *Clec11a[-/-]* and sex-matched littermate controls. Two weeks later, *Clec11a[-/-]* mice had significantly less callus bone around the fracture site (*Figure 4A*) and significantly more callus cartilage (*Figure 4B*) as compared to controls, suggesting delayed endochondral ossification. MicroCT analysis of the callus at the fracture site two weeks after the fracture revealed significantly reduced trabecular bone volume, trabecular number, trabecular thickness, and trabecular connectivity density (*Figure 4C–H*) and significantly increased trabecular spacing (*Figure 4I*) in *Clec11a[-/-]* bones. The bone mineral density in the callus did not

**Figure 3.** Clec11a is necessary for osteogenic differentiation. (A–D) Osteogenic differentiation in culture of bone marrow stromal cells from the femur bone marrow of *Clec11⁻ᐟ⁻* mice and sex-matched littermate controls. Alkaline phosphatase staining and alizarin red staining were performed after seven days (A and B) and 14 days (C and D) to quantify osteoblast differentiation and mineralization (n = 3 independent experiments). (E and F) Adipogenic differentiation in culture of bone marrow stromal cells from the femur bone marrow of *Clec11⁻ᐟ⁻* mice and sex-matched littermate controls. Oil red O staining was performed after four days (n = 3 independent experiments). (G and H) Chondrogenic differentiation in cell pellets of bone marrow stromal cells from the femur bone marrow of *Clec11⁻ᐟ⁻* mice and sex-matched littermate controls. Toluidine blue staining was performed on cryosectioned cell pellets after 21 days (n = 3 independent experiments). (I–K) Representative perilipin and osteopontin (OPN) staining in femur sections of two month-old *Clec11⁻ᐟ⁻* mice and sex-matched littermate controls (I and J) as well as the number of adipocytes per mm$^2$ in sections through the bone marrow metaphysis (K) (n = 3 mice per genotype, total, from three independent experiments). (L–N) Representative Safranin O/fast green staining in femur sections of two month-old *Clec11a⁻ᐟ⁻* mice and sex-matched littermate controls (L and M) as well as the number of chondrocytes per mm$^2$ in sections through the growth plate (N) (n = 3 mice per genotype from three independent experiments). The statistical significance of differences among genotypes was assessed using two-tailed Student's t tests. All data represent mean ± SD (**p<0.01, ***p<0.001) from female mice. The source data are in *Figure 3—source data 1*.

The following source data and figure supplements are available for figure 3:

**Source data 1.** Data for *Figure 3* and *Figure 3—figure supplements 1* and *2*.

**Figure supplement 1.** Analysis of osteoprogenitor populations and in vivo transplantation of bone marrow stromal cells.

*Figure 3 continued on next page*

*Figure 3 continued*

**Figure supplement 2.** Analysis of the number of adipocytes and chondrocytes in 10 month-old *Clec11a*$^{-/-}$ and control femur sections.

significantly differ between *Clec11a*$^{-/-}$ and control mice (*Figure 4J*). The callus volume, diameter, and polar moment of inertia were significantly increased in *Clec11a*$^{-/-}$ mice as compared to littermate controls (*Figure 4K–M*), further suggesting that fracture healing was compromised in *Clec11a*$^{-/-}$ mice (*O'Neill et al., 2012*).

## Clec11a promotes mesenchymal progenitor differentiation into mature osteoblasts

To test whether Clec11a is sufficient to promote osteogenesis we constructed a HEK293 cell line that stably expressed mouse Clec11a with a C-terminal Flag tag. We affinity purified recombinant Clec11a (rClec11a) from the culture medium using anti-Flag M2 beads. Wild-type bone marrow cells were cultured to form CFU-F, then replated and grown under osteogenic culture conditions. Addition of rClec11a to these cultures significantly increased alizarin red staining, suggesting increased mineralization (*Figure 5A and B*). Addition of rClec11a also rescued osteogenesis by *Clec11a*$^{-/-}$ bone marrow stromal cells at 7 and 14 days after induction of differentiation (*Figure 5—figure supplement 1J–M*). Transient expression of mouse *Clec11a* cDNA in the MC3T3-E1 mouse pre-osteoblast cell line (*Wang et al., 1999*) increased osteogenic differentiation by these cells in culture (*Figure 5—figure supplement 1H and I*).

To test whether Clec11a is sufficient to promote the differentiation of bone marrow mesenchymal progenitors into mature osteoblasts, we sorted LepR$^+$CD105$^+$CD45$^-$Ter119$^-$CD31$^-$ cells from wild-type mice into recombinant Clecl11a (rClec11a)-containing and control cultures at clonal density. Addition of rClec11a to these cultures did not significantly affect the percentage of cells that formed CFU-F or the number of cells per colony (*Figure 5—figure supplement 1A and B*). Upon induction of differentiation, rClec11a significantly increased the number of alkaline phosphatase positive osteoblasts per colony (*Figure 5—figure supplement 1D*). This appeared to reflect a promotion of differentiation as rClec11a did not significantly affect the frequencies of dividing osteoblasts (*Figure 5—figure supplement 1E*) or osteoblasts undergoing apoptosis within the colonies (*Figure 5—figure supplement 1F*). Consistent with the experiments above, addition of rClec11a did not significantly affect *Sp7*, *Runx2*, or *Col1a1* expression by cells within these cultures, but it did significantly increase the expression of *Ibsp* and *Dmp1* (*Figure 5—figure supplement 1G*). Clec11a is thus sufficient to promote the differentiation of bone marrow mesenchymal progenitors into mature osteoblasts.

To test whether rClec11a promotes osteogenesis in vivo, we administered daily subcutaneous injections of rClec11a to two month-old wild-type mice for 28 days. Consistent with the in vitro data, rClec11a dose-dependently increased trabecular bone volume in the distal femur metaphysis (*Figure 5C and D*). The higher doses of rClec11a also significantly increased trabecular number and reduced trabecular spacing (*Figure 5E–G*). The increased osteogenesis was associated with a significantly increased mineralized bone surface (*Figure 5H*) and bone formation rate (*Figure 5I*), but did not affect bone resorption (*Figure 5J*). Cortical bone parameters in the femur diaphysis were not affected by rClec11a in these experiments (*Figure 5—figure supplement 1N–S*). rClec11a thus promotes osteogenesis in wild-type mice in vivo.

To test whether administration of rClec11a can rescue the bone loss phenotype in *Clec11a*$^{-/-}$ mice, we administered daily subcutaneous injections of 50 μg/kg rClec11a to six month-old *Clec11a*$^{-/-}$ mice for 28 days. This restored plasma Clec11a to control levels (data not shown). Consistent with this, *Clec11a*$^{-/-}$ mice exhibited significantly increased trabecular bone volume (*Figure 5K and L*), trabecular number, trabecular thickness, connectivity density, and decreased trabecular spacing (*Figure 5M–Q*). After rClec11a administration to *Clec11a*$^{-/-}$ mice, trabecular bone parameters were similar to those in normal control mice.



**Figure 4.** Clec11a is necessary for bone regeneration and fracture healing. (A and B) Hematoxylin and eosin (A; dark blue = bone marrow cells; pink = bone) and Safranin O (B; red = cartilage) staining of the callus around the fracture site two weeks after bone fracture. F, fibrous tissue. BM, bone marrow. C, cartilage. (C and D) Representative microCT snapshot images of the callus (C) and cut-plane images around the fracture site (D) two weeks after bone fracture. (E–M) MicroCT analysis of trabecular bone volume/total volume (E), trabecular number (F), trabecular thickness (G), connectivity density (H), trabecular spacing (I), bone mineral density (J), callus volume (K), callus diameter (L) and polar moment of inertia (M) in the callus two weeks after bone fracture (n = 3 mice per genotype, total, from three independent experiments). The statistical significance of differences was assessed using two-tailed Student's t tests. All data represent mean ± SD (*p<0.05, **p<0.01) from male mice that were two months old at the time of fracture. The source data are in *Figure 4—source data 1*.

The following source data is available for figure 4:

**Source data 1.** Data for *Figure 4*.



**Figure 5.** Recombinant Clec11a promotes osteogenesis in vitro and in vivo. (**A** and **B**) Osteogenic differentiation of stromal cells from femur bone marrow of wild-type mice. Vehicle or 10 ng/ml rClec11a were added to osteogenic culture conditions and alizarin red staining was assessed 14 days later to test whether Clec11a would promote osteogenesis (n = 3 independent experiments with duplicate cultures per treatment per experiment). (**C**) Representative microCT images of trabecular bone in the distal femur metaphysis of wild-type female mice treated with daily subcutaneous doses of *Figure 5 continued on next page*

*Figure 5 continued*

rClec11a for 28 days (panels **C–G** reflect n = 6 mice per treatment, total, from six independent experiments). (**D–G**) MicroCT analysis of trabecular bone parameters from the distal femur metaphysis of mice treated with daily subcutaneous doses of rClec11a for 28 days. (**H** and **I**) Mineralized surface/bone surface ratio (**H**) and bone formation rate (**I**) in trabecular bone in the femur metaphysis of mice treated with rClec11a for 28 days (n = 3 mice per genotype, total, from three independent experiments). (**J**) Bone resorption analysis based on the deoxypyridinoline/creatinine ratio in the urine (n = 4 mice per genotype, total, from four independent experiments). (**K**) Representative microCT images of trabecular bone in the distal femur metaphysis of control mice treated with vehicle, or *Clec11a*$^{-/-}$ mice treated with vehicle or 50 µg/kg daily subcutaneous injections of rClec11a for 28 days (panels **K–Q** reflect n = 4–6 mice per treatment, total, from four independent experiments). (**L–Q**) Trabecular bone parameters from the distal femur metaphysis of mice in the experiment shown in panel **J**. The statistical significance of differences among treatments was assessed using one-way ANOVAs with Tukey's multiple comparison tests. All data represent mean ± SD (*p<0.05, **p<0.01, ***p<0.001) from female mice that were two months (**C–J**) or six months (**K–Q**) old at the start of the experiment. The source data are in *Figure 5—source data 1*.

The following source data and figure supplement are available for figure 5:

**Source data 1.** Data for *Figure 5* and *Figure 5—figure supplement 1*.
**Figure supplement 1.** Effects of rClec11a on osteogenic differentiation and cortical bone analysis in mice wild-type mice treated with rClec11a.

## rClec11a administration prevents osteoporosis

Ovariectomy in adult mice induces osteoporosis by increasing bone resorption (*Rodan and Martin, 2000*). We ovariectomized mice at two months of age then administered daily subcutaneous injections of recombinant human parathyroid hormone (PTH) fragment 1–34, rClec11a, or vehicle for 28 days before analysis by microCT. MicroCT analysis showed that trabecular bone volume and number were significantly reduced in ovariectomized mice (*Figure 6A–C*) while trabecular spacing was significantly increased (*Figure 6E*). Daily administration of PTH to ovariectomized mice significantly increased trabecular bone volume (*Figure 6B*) and trabecular number (*Figure 6C*), while reducing trabecular spacing (*Figure 6E*). Ovariectomy did not significantly affect the plasma Clec11a level (data not shown). Daily administration of rClec11a to ovariectomized mice significantly increased trabecular bone volume (*Figure 6B*) and trabecular number (*Figure 6C*), while reducing trabecular spacing (*Figure 6E*). Cortical bone parameters were not significantly changed by PTH or Clec11a administration in these experiments (*Figure 6—figure supplement 1A–F*). rClec11a can therefore prevent the loss of trabecular bone in ovariectomized mice.

Consistent with the fact that ovariectomy increases bone resorption (*Harada and Rodan, 2003*), the urinary bone resorption marker deoxypyridinoline was significantly increased in ovariectomized mice as compared to sham operated controls (*Figure 6F*). Administration of rClec11a or PTH did not significantly affect deoxypyridinoline levels (*Figure 6F*) or numbers of osteoclasts (*Figure 6—figure supplement 1G*) in ovariectomized mice. However, based on calcein double labeling and histomorphometry analysis, the trabecular bone formation rate (*Figure 6G*) and the number of osteoblasts associated with trabecular bones (*Figure 6—figure supplement 1H*) were significantly increased by rClec11a or PTH administration. rClec11a thus prevented the loss of trabecular bone in ovariectomized mice by promoting bone formation.

We also assessed the effect of rClec11a on a model of secondary osteoporosis in which bone loss was induced by dexamethasone injection, mimicking glucocorticoid-induced osteoporosis in humans (*McLaughlin et al., 2002*; *Weinstein et al., 1998*). Daily intraperitoneal administration of 20 mg/kg dexamethasone for four weeks in mice significantly reduced lymphocyte numbers in the blood without significantly affecting neutrophil or monocyte counts (*Figure 6—figure supplement 2A–D*). MicroCT analysis of the distal femur metaphysis showed significantly reduced trabecular bone volume and thickness in the dexamethasone-treated as compared to vehicle-treated mice (*Figure 6H–L*). Treatment of dexamethasone-treated mice with PTH significantly increased trabecular bone volume, trabecular number, and trabecular thickness while significantly reducing trabecular spacing (*Figure 6H–L*). Dexamethasone treatment did not significantly affect plasma Clec11a levels (*Figure 6—figure supplement 2K*), but administration of rClec11a to dexamethasone-treated mice significantly increased trabecular bone volume and trabecular number while significantly reducing trabecular spacing (*Figure 6H–L*). Dexamethasone treatment also significantly reduced cortical

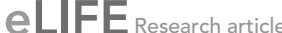

**Figure 6.** Recombinant Clec11a prevents osteoporosis. (**A**) Representative microCT images of trabecular bone in the distal femur metaphysis. Two month-old sham operated (Mock) or ovariectomized (OVX) female mice received daily subcutaneous injections with vehicle, 40 μg/kg human PTH, or 50 μg/kg rClec11a for 28 days. (**B–E**) MicroCT analysis of trabecular bone parameters in the distal femur metaphysis of the mice from the experiment in panel A (n = 6–8 mice per treatment, total, from six independent experiments). (**F**) Bone resorption analysis based on the deoxypyridinoline/creatinine

*Figure 6 continued on next page*

*Figure 6 continued*

ratio in the urine (n = 4 mice per treatment, total, from four independent experiments). (**G**) Trabecular bone formation rate based on calcium double labeling in the distal femur metaphysis (n = 3–4 mice per treatment, total, from at least three independent experiments). (**H**) Representative microCT images of trabecular bone in the distal femur metaphysis. Two month-old wild-type female mice were treated with daily intraperitoneal injections of PBS or 20 mg/kg dexamethasone (DEX) for 28 days, with or without daily subcutaneous injections of vehicle, 40 µg/kg human PTH, or 50 µg/kg rClec11a. (**I–L**) Trabecular bone parameters of mice from the same experiments (panels I–N reflect n = 4 mice per treatment, total, from four independent experiments). (**M**) Trabecular bone formation rate based on calcium double labeling in the distal femur metaphysis (n = 3–4 mice per treatment, total, from at least three independent experiments). (**N**) Bone resorption analysis based on the deoxypyridinoline/creatinine ratio in the urine (n = 4 mice per treatment, total, from four independent experiments). The statistical significance of differences was assessed using one-way ANOVAs with Tukey's multiple comparisons tests. All data represent mean ± SD (*p<0.05, **p<0.01, ***p<0.001) from female mice that were two months old at the start of the experiment. The source data are in *Figure 6—source data 1*.
The following source data and figure supplements are available for figure 6:

**Source data 1.** Data for *Figure 6* and *Figure 6—figure supplement 1* and *2*.
**Figure supplement 1.** Cortical bone analysis in ovariectomized mice.
**Figure supplement 2.** Hematopoietic and cortical bone analysis in dexamethasone-injected mice.

thickness but neither PTH nor rClec11a rescued this effect in these experiments (*Figure 6—figure supplement 2E–J*).

Consistent with the fact that dexamethasone reduces bone formation (*Harada and Rodan, 2003*), the rate of trabecular bone formation based on calcein double labeling (*Figure 6M*) and the numbers of osteoblasts in trabecular bone (*Figure 6—figure supplement 2L*) were significantly reduced in dexamethasone-treated as compared to vehicle-treated mice. Administration of PTH or rClec11a significantly increased the trabecular bone formation rate (*Figure 6M*) and the number of osteoblasts (*Figure 6—figure supplement 2L*) in dexamethasone-treated mice. As expected, dexamethasone treatment, or administration of PTH or rClec11a, did not significantly affect deoxypyridinoline levels (*Figure 6N*) or osteoclast numbers (*Figure 6—figure supplement 2M*). rClec11a thus prevented the loss of trabecular bone in dexamethasone-treated mice by promoting bone formation.

## rClec11a administration reverses osteoporosis

To test whether rClec11a could reverse bone loss after the onset of osteoporosis we ovariectomized two month-old wild-type mice and waited for four weeks before administering PTH or rClec11a daily for another four weeks. MicroCT analysis showed that trabecular and cortical bone volumes as well as trabecular number were significantly reduced in ovariectomized mice (*Figure 7A–C*, and *Figure 7—figure supplement 1C*). Trabecular spacing was significantly increased in ovariectomized mice (*Figure 7E*). Daily administration of PTH to ovariectomized mice increased trabecular bone volume (*Figure 7B*) and significantly reduced trabecular spacing relative to untreated ovariectomized mice (*Figure 7E*). PTH treatment also significantly increased cortical area (*Figure 7—figure supplement 1C*) and cortical thickness (*Figure 7—figure supplement 1E*). Daily administration of rClec11a to ovariectomized mice significantly increased trabecular bone volume (*Figure 7B*) and trabecular number (*Figure 7C*), while reducing trabecular spacing (*Figure 7E*) relative to untreated ovariectomized mice. rClec11a did not significantly affect cortical area (*Figure 7—figure supplement 1C*) or cortical thickness (*Figure 7—figure supplement 1E*) in ovariectomized mice. rClec11a can thus reverse trabecular bone loss after the onset of ovariectomy-induced osteoporosis.

## Recombinant human Clec11a promotes osteogenesis

To test whether human Clec11a also promotes osteogenesis, we constructed a HEK293 cell line that stably expressed human Clec11a with a C-terminal Flag tag, then affinity purified recombinant human Clec11a (rhClec11a) from the culture medium. Addition of rhClec11a to human bone marrow stromal cells cultured under osteogenic culture conditions significantly increased osteoblast differentiation based on alkaline phosphatase (*Figure 7H and I*) and alizarin red staining (*Figure 7J and K*).

**Figure 7.** Recombinant Clec11a reverses osteoporosis and promotes osteogenesis by human bone marrow stromal cells. (**A**) Representative microCT images of trabecular bone in the distal femur metaphysis. Two month-old sham operated (Mock) or ovariectomized (OVX) female mice were left untreated for 28 days for osteoporosis to develop, and then received daily subcutaneous injections with vehicle, 40 µg/kg human PTH, or 50 µg/kg rClec11a for another 28 days. (**B–G**) MicroCT analysis of trabecular bone parameters in the distal femur metaphysis of the mice from the experiment in

*Figure 7 continued on next page*

*Figure 7 continued*

panel **A** (panels **B–G** reflect n = 4 mice per treatment, total, from four independent experiments). The statistical significance of differences among treatments was assessed using one-way ANOVAs with Tukey's multiple comparisons tests. All data represent mean ± SD (*p<0.05, **p<0.01, ***p<0.001) from female mice that were two months old at the start of the experiment. (**H–K**) Osteogenic differentiation in culture of hMSCs. Alkaline phosphatase staining and alizarin red staining were performed after eight days (**H** and **I**) and 21 days (**J** and **K**) to quantify osteoblast differentiation and mineralization (n = 6 independent experiments). (**L–O**) Ossicle formation by human bone marrow stromal cells in NSG mice. Vehicle or 50 μg/kg rhClec11a was subcutaneously injected daily for 4 (**L** and **M**) or 8 (**N** and **O**) weeks before the ossicles were dissected and sectioned for H and E staining (n = 5–9 ossicles per treatment, total, from two independent experiments). F, fibrous tissue. HA, HA/TCP carrier. The statistical significance of differences in **H–O** was assessed using two-tailed Student's t tests. Data represent mean ± SD (*p<0.05 **p<0.01, ***p<0.001). The source data are in *Figure 7—source data 1*.

The following source data and figure supplement are available for figure 7:

**Source data 1.** Data for *Figure 7* and *Figure 7—figure supplement 1*.
**Figure supplement 1.** Cortical bone analysis in ovariectomized mice.

To test whether rhClec11a promotes osteogenesis by human bone marrow stromal cells in vivo we subcutaneously transplanted a suspension of human bone marrow stromal cells, hydroxyapatite/tricalcium phosphate particles, and fibrin gel into immunocompromised NSG mice (*Bianco et al., 2006*) and administered daily subcutaneous injections of rhClec11a or vehicle for 4 or 8 weeks. rhClec11a significantly accelerated bone formation in the ossicles after four weeks and significantly increased bone formation in the ossicles after eight weeks (*Figure 7L–O*). rhClec11a thus promotes osteogenesis by human bone marrow stromal cells in vivo.

## Discussion

We have identified a new osteogenic factor, Clec11a, which maintains the adult skeleton by promoting the osteogenesis. Clec11a was necessary and sufficient to promote osteogenesis in culture and in vivo. *Clec11a* deficiency significantly reduced bone volume in both limb bones and vertebrae of adult mice (*Figure 2*). Clec11a was expressed by subsets of bone marrow stromal cells, osteoblasts, osteocytes, and hypertrophic chondrocytes, particularly in the metaphysis, but also in portions of cortical bone (*Figure 1*). In light of the unanticipated osteogenic activity of Clec11a, and its role in the maintenance of the adult skeleton, we propose to call this growth factor Osteolectin, a name that is more descriptive of both biological function and protein structure.

Clec11a/Osteolectin appears to promote bone formation by promoting the differentiation of mesenchymal progenitors into mature osteoblasts. *Clec11a/Osteolectin* deficient bone marrow stromal cells formed significantly fewer osteoblasts and mineralized bone matrix in culture (*Figure 3A–D*) with significantly lower expression of the mature osteoblast markers *Ibsp* and *Dmp1* (*Figure 3—figure supplement 1N*). Addition of recombinant Clec11a/Osteolectin to cultures of wild-type bone marrow stromal cells significantly increased the formation of osteoblasts (*Figure 5—figure supplement 1D*) as well as *Ibsp* and *Dmp1* expression (*Figure 5—figure supplement 1G*). Clec11a/Osteolectin deficient bone marrow stromal cells also formed less bone in the ossicles in vivo (*Figure 3—figure supplement 1O–Q*). Recombinant Clec11a/Osteolectin promoted bone formation in the ossicles in vivo (*Figure 7L–O*). In contrast to its effects on osteogenic differentiation, *Clec11a/Osteolectin* deficiency did not affect the frequency, proliferation, or survival of bone marrow mesenchymal progenitors in vivo (*Figure 3—figure supplement 1B–M*). Recombinant Clec11a/Osteolectin also did not have any effect on the frequency of CFU-F that formed colonies in culture (*Figure 5—figure supplement 1A*) or the number of cells per colony (*Figure 5—figure supplement 1B*). We conclude that Clec11a/Osteolectin promotes the osteogenic differentiation of mesenchymal progenitors but not their proliferation or survival.

Hypertrophic chondrocytes can also transdifferentiate into osteoblasts and osteocytes during endochondral ossification (*Ono et al., 2014b*; *Yang et al., 2014*; *Zhou et al., 2014b*). Thus, in addition to promoting the differentiation of mesenchymal progenitors into mature osteoblasts, Clec11a/

Osteolectin might also promote the transdifferentiation of hypertrophic chondrocytes into osteoblasts. Lineage tracing studies will be required in future to test this.

Nobody has yet identified the receptor for Clec11a/Osteolectin, limiting our ability to study the signaling mechanisms by which it promotes osteogenesis. We do observe binding of flag-tagged Clec11a/Osteolectin to the surface of osteogenic cell lines (data not shown). We hypothesize that Clec11a/Osteolectin promotes the osteogenic differentiation of mesenchymal progenitors by binding to a signaling receptor on the surface of these cells. Identification of this receptor will require significant additional work beyond the scope of the current study.

Phylogenic analysis showed that Clec11a/Osteolectin is most closely related to Clec3b/Tetranectin. *Tetranectin* expression increases during mineralization by osteogenic progenitors in culture and overexpression of *Tetranectin* in PC12 cells increases the bone content of tumors formed by these cells (*Wewer et al., 1994*). Tetranectin deficient mice exhibit kyphosis as a result of asymmetric growth plate development in vertebrae (*Iba et al., 2001*); however, it is unknown whether Tetranectin is required for osteogenesis in vivo. Tetranectin is found in both cartilaginous fish and bony fish but Clec11a/Osteolectin is only found in bony fish and higher vertebrate species. This suggests that Clec11a/Osteolectin evolved in bony species to promote osteogenic differentiation and mineralization. Among mammals Clec11a/Osteolectin is highly conserved: human and mouse Clec11a/Osteolectin proteins are 85% identical and 90% similar. Consistent with this, recombinant human Clec11a/Osteolectin promoted osteogenesis by human bone marrow stromal cells in culture and in vivo (*Figure 7H–O*).

## Materials and methods

### Mice and cell lines

To generate *Clec11a$^{-/-}$* mice, *Cas9* mRNA and *sgRNAs* were transcribed using mMESSAGE mMACHINE T7 Ultra Kit and MEGAshortscript Kit (Ambion), purified by MEGAclear Kit (Ambion), and microinjected into C57BL/6 zygotes by the Transgenic Core Facility of the University of Texas Southwestern Medical Center (UTSW). Chimeric mice were genotyped by restriction fragment length polymorphism (RFLP) analysis and backcrossed onto a C57BL/Ka background to obtain germline transmission. Mutant mice were backcrossed onto a C57BL/Ka background for 3 to 6 generations prior to analysis. Wild-type C57BL/Ka mice were used for rClec11a injection, ovariectomy, and dexamethasone injection experiments. All procedures were approved by the UTSW Institutional Animal Care and Use Committee (Animal protocol number: 2016–101334-G). The cell lines used in this study included HEK293 and MC3T3-E1 (Subclone 4), which were obtained from ATCC and authenticated by STR profiling. They were shown to be free of mycoplasma contamination.

### Flow cytometry

Antibodies used to analyze hematopoietic stem cells (HSCs) and multipotent hematopoietic progenitors (MPPs) included anti-CD150-PE-Cy5 (BioLegend, clone TC15-12F12.2, 1:200), anti-CD48-FITC (eBioscience, clone HM48-1, 1:200), anti-Sca-1-PEcy7 (eBioscience, E13-161.7, 1:200), anti-c-Kit-APC-eFluor780 (eBioscience, clone 2B8, 1:200) and the following antibodies against lineage markers: anti-Ter119-PE (eBioscience, clone TER-119, 1:200), anti-B220-PE (BioLegend, clone 6B2, 1:400), anti-Gr-1-PE (BioLegend, clone 8C5, 1:800), anti-CD2-PE (eBioscience, clone RM2-5, 1:200), anti-CD3-PE (BioLegend, clone 17A2, 1:200), anti-CD5-PE (BioLegend, clone 53–7.3, 1:400) and anti-CD8-PE (eBioscience, clone 53–6.7, 1:400). The following antibodies were used to identify restricted hematopoietic progenitors: anti-CD34-FITC (eBioscience, clone RAM34, 1:100); anti-CD16/32 Alexa Fluor 700 (eBioscience, clone 93, 1:200); anti-CD135-PEcy5 (eBioscience, clone A2F10, 1:100); anti-CD127-Biotin (BioLegend, clone A7R34, 1:200) + Streptavidin-PE-CF592 (BD Biosciences, 1:500); anti-cKit-APC-eFluor780 (eBioscience, clone 2B8, 1:200); anti-Sca1-PEcy7 (eBioscience, clone E13-161.7, 1:200) and lineage markers listed above. The following antibodies were used to identify differentiated cells: anti-CD71-FITC (BD Biosciences, clone C2, 1:200); anti-Ter119-APC (eBioscience, clone TER-119, 1:200); anti-CD3-PE (BioLegend, clone 17A2, 1:200); anti-B220-PEcy5 (eBioscience, clone RA3-6B2, 1:400); anti-Mac-1-APC-eFluor780 (eBioscience, M1/70, 1:200) and anti-Gr-1-PEcy7 (BioLegend, clone RB6-8C5, 1:400). Anti-CD45.2-FITC (BioLegend, clone 104, 1:200) and anti-CD45.1-APC-eFluor-78 (eBioscience, clone A20, 1:100) were used to distinguish donor from

recipient cells in competitive reconstitution assays. The following antibodies were used to distinguish subpopulations of bone marrow stromal cells: anti-CD45-APC (eBioscience, clone 30-F11, 1:200), anti-Ter119-APC (eBioscience, clone TER-119, 1:200), anti-CD31-APC (Biolegend, clone MEC13.3, 1:200), anti-Tie2-APC (BioLegend, clone TEK4, 1:200), anti-Thy1.1-FITC (eBioscience, clone HIS51, 1:200), anti-Thy1.2-FITC (eBioscience, clone 30-H12, 1:200), anti-CD51-biotin (BioLegend, clone RMV-7, 1:100), anti-CD51-PE (BioLegend, clone RMV-7, 1:100), anti-Ly-51-PEcy7 (BioLegend, clone 6C3, 1:200), anti-CD200-PE (BioLegend, clone OX-90, 1:200), anti-CD105-Pacific Blue (BioLegend, clone MJ7/18), anti-LepR-biotin antibody (R and D systems, BAF497, 1:100) and anti-PDGFRα-biotin (eBioscience, clone APA5, 1:200). Cells were stained with antibodies in 200 µl of staining medium (HBSS + 2% fetal bovine serum) on ice for 1 hr, and then washed by adding 2 ml of staining buffer followed by centrifugation. Biotin-conjugated antibodies were incubated with streptavidin-PE or streptavidin-Brilliant Violet 421 (Biolegend, 1:500) for another 20 min (Biolegend, 1:500). Cells were resuspended in staining medium with 1 µg/ml DAPI (Invitrogen) and analyzed with a FACSCanto flow cytometer (BD Biosciences) or sorted using a FACSAria flow cytometer (BD Biosciences) with a 130 µm nozzle. To assess proliferation in vivo, mice were given a single intraperitoneal injection of BrdU (100 mg/kg body mass) and maintained on 0.5 mg/ml BrdU in the drinking water for 14 days. The frequency of BrdU+ SSCs was then analyzed by flow cytometry using the APC BrdU Flow Kit (BD Biosciences).

## Bone marrow digestion and CFU-F culture

Enzymatic digestion of bone marrow cells and CFU-F cultures were performed as described previously (*Suire et al., 2012*). Briefly, intact marrow plugs were flushed from the long bones and subjected to two rounds of enzymatic digestion at 37°C for 15 min each. The digestion buffer contained 3 mg/ml type I collagenase (Worthington), 4 mg/ml dispase (Roche Diagnostic) and 1 U/ml DNase I (Sigma) in HBSS with calcium and magnesium. The cells were resuspended in staining medium (HBSS + 2% fetal bovine serum) with 2 mM EDTA to stop the digestion. To form CFU-F colonies, freshly dissociated single-cell suspensions were plated at clonal density in 6-well plates ($5 \times 10^5$ cells/well) or 10 cm plates ($5 \times 10^6$ cells/dish) with DMEM (Gibco) plus 20% fetal bovine serum (Sigma F2442, lot 14M255, selected to support CFU-F growth), 10 µM ROCK inhibitor (Y-27632, TOCRIS), and 1% penicillin/streptomycin (Invitrogen). Cultures were maintained at 37°C in gas-tight chambers (Billups-Rothenberg, Del Mar, CA) that were flushed daily for 30 s with 5% $O_2$ and 5% $CO_2$ (balance Nitrogen) to maintain a low oxygen environment that promoted survival and proliferation (*Morrison et al., 2000*). The CFU-F culture medium was changed on the second day after plating to wash out contaminating macrophages, then changed every 3–4 days after that. CFU-F colonies were counted eight days after plating by staining with 0.1% Toluidine blue in 4% formalin solution.

## In vitro differentiation

Osteogenic and adipogenic differentiation were assessed by replating primary CFU-F cells into 48-well plates (25,000 cells/cm²). On the second day of culture, the medium was replaced with adipogenic (four days) or osteogenic (seven days or 14 days) medium (StemPro MSC differentiation kits; Life Technologies). Equal numbers of cells from wild-type and *Clec11a⁻/⁻* cultures were replated so there was no difference in the density of cells of different genotypes. In some experiments, clonal differentiation potential was assessed by sorting 500 bone marrow LepR+CD105+-CD45⁻Ter119⁻CD31⁻ cells into each well of a 6-well plate to form CFU-F colonies at clonal density over an eight day period. Then the culture medium was replaced with osteogenic differentiation medium. Seven to 14 days later, the percentage of colonies that contained osteoblasts and the numbers of osteoblasts per colony were quantified by StemTAG Alkaline Phosphatase Staining and Activity Assay Kit (Cell Biolabs) and alizarin red staining (Sigma).

Chondrogenic potential was assessed by centrifuging $2 \times 10^5$ CFU-F cells to form cell pellets, which were then cultured in chondrogenic medium for 21 days (StemPro chondrogenesis differentiation kit; Life Technologies), changing the culture medium every 2–3 days. Chondrocyte formation within the cell pellets was assessed by cryosectioning and Toluidine blue staining (*Robey et al., 2014*). The osteogenic differentiation of human bone marrow stromal cells and MC3T3-E1 cells was

tested using the StemPro osteogenesis differentiation kit (Life Technologies). Images were acquired using an Olympus IX81 microscope.

## MicroCT analysis

Femurs and lumbar vertebrae were dissected, fixed overnight in 4% paraformaldehyde and stored in 70% ethanol at 4°C. Femurs and lumbar vertebrae were scanned at an isotropic voxel size of 3.5 μm and 7 μm, respectively, with peak tube voltage of 55 kV and current of 0.145 mA (μCT 35; Scanco Medical AG, Bassersdorf, Switzerland). A three-dimensional Gaussian filter (σ = 0.8) with a limited, finite filter support of one was used to suppress noise in the images, and a threshold of 263–1000 was used to segment mineralized bone from air and soft tissues. Trabecular bone parameters were measured in the distal metaphysis of the femurs. The region of interest was selected from below the distal growth plate where the epiphyseal cap structure completely disappeared and continued for 100 slices toward the proximal end of the femur. Contours were drawn manually a few voxels away from the endocortical surface to define trabecular bones in the metaphysis (*Bouxsein et al., 2010*). Cortical bone parameters were measured by analyzing 100 slices in mid-diaphysis femurs. Vertebral bone parameters were measured by analyzing 200 slices in the middle of L3 lumbar vertebrae. The fracture callus was scanned at an isotropic voxel size of 6 μm with the same settings as described above. The region of interested was selected from 200 slices above the fracture site to 200 slices below the fracture site (400 slices in total), including the entire callus. A segmentation threshold of 263–500 was used to analyze the trabecular parameters in the fracture callus.

## Long-term competitive reconstitution assays in irradiated mice

Two month-old adult recipient mice were irradiated with an XRAD 320 irradiator (Precision X-Ray Inc.), giving two doses of 550 rad, delivered at least 2 hr apart. C57BL/Ka-Thy-1.1 (CD45.2) donor mice and C57BL/Ka-Thy-1.2(CD45.1) recipient mice were used in transplant experiments. 300,000 donor whole bone marrow cells from *Clec11a*$^{-/-}$ or littermate control mice (CD45.2) were transplanted along with 300,000 recipient whole bone marrow cells (CD45.1) into lethally irradiated recipient mice (C57BL/Ka-Thy-1.1 x C57BL/Ka-Thy-1.2 (CD45.1/CD45.2) heterozygotes). Peripheral blood was obtained from the tail veins of recipient mice at 4 to 16 weeks after transplantation. Blood was subjected to ammonium-chloride lysis of the red blood cells and leukocytes were stained with antibodies against CD45.1, CD45.2, B220, Mac-1, CD3 and Gr-1 to assess hematopoietic chimerism by donor and recipient cells by flow cytometry.

## Bone sectioning and immunostaining

Dissected bones were fixed in 4% paraformaldehyde overnight, decalcified in 10% EDTA for four days, and dehydrated in 30% sucrose for two days. Bones were sectioned (10 μm) using the Cryo-Jane tape-transfer system (Leica). Sections were blocked in PBS with 10% horse serum for 30 min and then stained overnight at 4°C with goat IgG control (R and D Systems, 1:500), goat anti-Clec11a antibody (R and D systems, 1:500), rabbit anti-Aggrecan antibody (Chemicon, 1:500), rabbit anti-Perilipin antibody (Sigma, 1:2000) or goat anti-Osteopontin antibody (R and D, 1:500). Donkey anti-goat Alexa Fluor 488, Donkey anti-goat Alexa Fluor 647 and donkey anti-rabbit Alexia Fluor 555 were used as secondary antibodies (Invitrogen, 1:500). Slides were mounted with anti-fade prolong gold with DAPI (Invitrogen). Images were acquired using a Zeiss LSM780 confocal microscope or Olympus IX81 microscope.

## Calcein double labeling and histomorphometry analysis

On day 0 and day 7, mice were injected intraperitoneally with 10 mg/kg body mass calcein dissolved in calcein buffer (0.15 M NaCl plus 2% NaHCO$_3$ in water) and sacrificed on day 9. The tibias were fixed overnight in 4% paraformaldehyde at 4°C, dehydrated in 30% sucrose for two days and sectioned without decalcification (7 μm sections). Mineral apposition and bone formation rates were determined as previously described (*Egan et al., 2012*). For the quantification of osteoblast number/bone surface and osteoclast number/bone surface, decalcified 10 μm femur sections were stained histochemically for alkaline phosphatase (Roche) or tartrate-resistant acid phosphatase (Sigma) activity. Growth plate chondrocytes were identified based on staining with Safranin O/fast

green (American MasterTech) and quantified using Image J. Alizarin red/alcian blue double staining was performed as described previously (*Ovchinnikov, 2009*).

## Bone fractures

A stainless steel wire was inserted into the intramedullary canal of the femur through the knee after anesthesia, and a bone fracture was introduced in the femur mid-diaphysis by 3-point bending. Buprenorphine was injected every 12 hr up to 72 hr after the surgery.

## Bone resorption analysis

Bone resorption rate was determined by measuring urinary levels of deoxypyridinoline (DPD) using a MicroVue DPD ELISA Kit (Quidel). The DPD values were normalized to urinary creatinine levels using the MicroVue Creatinine Assay Kit (Quidel).

## Recombinant protein purification

Mouse Clec11a cDNA was cloned into pcDNA3 vector (Invitrogen) containing a C-terminal 1XFlag-tag, which was then transfected into HEK293 cells with Lipofectamine 2000 (Invitrogen) and subjected to stable cell line selection using 1 mg/ml G418 (Sigma). Stable clones with high Clec11a expression were cultured in DMEM plus 10% FBS (Sigma), and 1% penicillin/streptomycin (Invitrogen). Culture medium was collected every two days, centrifuged to eliminate cellular debris, and stored with 1 mM phenylmethylsulfonyl fluoride at 4°C to inhibit protease activity. One litter of culture medium was filtered through a 0.2 µm membrane to eliminate cellular debris (Nalgene) before being loaded onto a chromatography column containing 2 ml Anti-FLAG M2 Affinity Gel (Sigma), with a flow rate of 1 ml/min. The column was sequentially washed using 20 ml of high salt buffer (20 mM Tris-HCl, 300 mM KCl, 10% Glycerol, 0.2 mM EDTA) followed by 20 ml of low salt buffer (20 mM Tris-HCl, 150 mM KCl, 10% Glycerol, 0.2 mM EDTA) and finally 20 ml of PBS. The FLAG-tagged Clec11a was then eluted from the column using 10 ml 3X FLAG peptide (100 µg/ml) in PBS or protein storage buffer (50 mM HEPES, 150 mM NaCl and 10% glycerol, pH = 7.5). Eluted protein was concentrated using Amicon Ultra-15 Centrifugal Filter Units (Ultracel-10K, Millipore), then quantitated by SDS-PAGE and colloidal blue staining (Invitrogen) and stored at −80°C. The recombinant human Clec11a was generated and purified in the same way.

## Osteoporosis models

For ovariectomy-induced osteoporosis, 8 week-old virgin female mice were anesthetized using Isoflurane, shaved, and disinfected with Betadine. A dorsal midline incision was made and the periovarian fat pad was gently grasped to exteriorize the ovary. The fallopian tube was then clamped off and the ovary was removed by cutting above the clamped area. The uterine horn was returned into the abdomen and the same process was repeated on the other side. After surgery, buprenorphine was given for analgesia, and mice were closely monitored until they resumed full activity. Vehicle, 40 µg/kg PTH (1–34) or 50 µg/kg rClec11a were subcutaneously injected daily starting one day or four weeks after the surgery and continuing for 28 days then the mice were analyzed. For dexamethasone-induced osteoporosis, PBS or 20 mg/kg dexamethasone was injected peritoneally into eight week-old virgin female mice daily for 28 days. Vehicle, 40 µg/kg PTH (1–34) or 50 µg/kg rClec11a were subcutaneously injected at the same time.

## In vivo transplantation of bone marrow stromal cells

Mouse and human bone marrow stromal cell ossicle formation in vivo was assessed as described previously (*Bianco et al., 2006*). Briefly, $2 \times 10^6$ mouse primary CFU-F cells were seeded into collagen sponges (Gelfoam, Pfizer), incubated at 37°C for 90 min, with or without 10 ng/ml rClec11a, and then transplanted subcutaneously into NSG mice. The ossicles formed by these cells were analyzed eight weeks after transplantation by cryosectioning and immunostaining with antibodies against perilipin and osteopontin. For human bone marrow stromal cells, $2 \times 10^6$ cells were incubated with 40 mg of hydroxyapatite (HA)/tricalcium phosphate (TCP) particles (65%/35%, Zimmer Dental, Warsaw IN), with or without 10 ng/ml rhClec11a, and rotated for 2 hr at 37°C. The cell/carrier slurry was centrifuged at 135xg for 5 min and embedded in a fibrin gel by adding 15 µl of human fibrinogen (3.2 mg/ml in sterile PBS) with 15 µl of human thrombin (25 U/ml in sterile 2% $CaCl_2$ in PBS). The gels

were left at room temperature for 10 min to clot, then transplanted subcutaneously into NSG mice. The ossicles formed by these cells were analyzed 4 or 8 weeks after transplantation by cryosectioning and H and E staining. Some of the mice were treated with daily subcutaneous injections of 50 µg/kg human recombinant Clec11a.

## qPCR

For quantitative reverse transcription PCR (qPCR), 6000 PDGFRα$^+$CD45$^-$Ter119$^-$CD31$^-$ cells were flow cytometrically sorted from enzymatically dissociated bone marrow into Trizol (Invitrogen). RNA was extracted and reverse transcribed into cDNA using SuperScript III (Invitrogen). qPCR was performed using a Roche LightCycler 480. The primers used for qPCR analysis included *Clec11a (NM_009131.3)*: 5'-AGG TCC TGG GAG GGA GTG-3' and 5'-GGG CCT CCT GGA GAT TCT T-3'; *Runx2 (NM_001146038.2)*: 5'- TTA CCT ACA CCC CGC CAG TC-3' and 5'-TGC TGG TCT GGA AGG GTC C-3'; *Sp7 (NM_130458.3)*: 5'- ATG GCG TCC TCT CTG CTT GA-3' and 5'-GAA GGG TGG GTA GTC ATT TG-3'; *Ibsp (NM_008318.3)*: 5'-AGT TAG CGG CAC TCC AAC TG-3' and 5'-TCG CTT TCC TTC ACT TTT GG-3'; *Dmp1* (NM_016779.2): 5'-TGG GAG CCA GAG AGG GTA G-3' and 5'- TTG TGG TAT CTG GCA ACT GG-3'; *Actb (NM_007393.5)*: 5'-GCT CTT TTC CAG CCT TCC TT-3' (Forward) and 5'-CTT CTG CAT CCT GTC AGC AA-3' (Reverse).

## EdU incorporation and caspase-3/7 activity

EdU was added into osteogenic differentiation medium at day 0 (10 µM final concentration) and maintained for the duration of the differentiation phase of the culture (8 days). The cultures were fixed by adding 1% paraformaldehyde on ice for 5 min then stained with alkaline phosphatase substrates (NBT/BCIP, Roche). Cells were then incubated with PBS supplemented with 3% FCS and 0.1% saponin for 5 min at room temperature, followed by Click-iT Plus reaction cocktail (Life Technologies) incubated for 30 min with 5 µM Alexa Fluor 555-azide. Cells were washed with PBS supplemented with 3% FCS and 0.1% saponin twice and quantified using an Olympus IX81 microscope. Caspase-3/7 enzymatic activity within individual cells growing adherently in culture plates was measured by adding CellEvent Caspase-3/7 Green Detection Reagent (a substrate for activated caspase-3/7, 2 µM final concentration; Life Technologies) to the differentiation medium at the end of the experiment and incubated for 30 min before fixation, alkaline phosphatase staining, and quantification.

## Blood cell counts

Peripheral blood was collected from the tail vein using Microvette CB 300 K2E tubes (Sarstedt) and counted using a HEMAVET HV950 cell counter (Drew Scientific).

## Hematopoietic colony formation

Hematopoietic colony formation was assessed by seeding 20,000 unfractionated mouse femur bone marrow cells into MethoCult M3334 or MethoCult M3234 supplemented with 10 ng/ml GM-CSF (STEMCELL Technologies). The cultures were incubated at 37°C for 10 days and then colonies were counted under the microscope.

## ELISA measurement of plasma Clec11a level

Mouse plasma was diluted 1:1 using 2x PBS buffer and 100 µl of diluted serum was coated on each well of the 96-well ELISA plate (COSTAR 96-WELL EIA/RIA STRIPWELL PLATE) at 4°C for 16 hr. The plate was then washed three times with washing buffer (PBS with 0.1% Tween-20), blocked with 300 µl ELISA Blocker Blocking Buffer (Thermo, N502) for 2 hr at room temperature, and washed for three times with washing buffer. Anti-Clec11a antibody (1 µg/ml diluted in 100 µl of PBS buffer with 0.1% Tween-20 per well) was then added and incubated at room temperature for 2 hr, washed three times with washing buffer, followed by HRP-conjugated donkey anti-Goat IgG secondary antibody (0.8 µg/ml diluted in 100 µl of PBS buffer with 0.1% Tween-20 per well) incubation at room temperature for 1 hr. After washing for three times, 100 µl of SureBlue TMB Microwell Peroxidase Substrate was added to each well and incubated at room temperature in the dark for 15 min. Finally, 100 µl of the TMB stop solution was added into each well and the optical density was measured at 450 nm.

## Biomechanical analysis

To assess biomechanical properties, the right femurs were harvested from mice, wrapped in saline-soaked gauze, and stored at –20°C. The femurs were rehydrated in PBS for at least 3 hr before testing, and then kept in a humidified chamber and preconditioned with 20 cycles of bending displacement (0.1 mm). Without immersion, the femurs were loaded to fracture with three-point bending (each holding point was 4 mm from the middle break point) under displacement control (3 mm/min) using a material testing system (Instron model # 5565, Norwood, MA).

## Genotyping

To genotype *Clec11a*$^{+/+}$, *Clec11*$^{+/-}$, and *Clec11a*$^{-/-}$ mice the following primers were used: 5'-TTT GGG TGC TGG GAA GCC C-3' and 5'-TTG CAC TGA GTC GCG GGT G-3' (*Clec11a*$^{+/+}$: 910 bp; *Clec11*$^{+/-}$ or *Clec11a*$^{-/-}$: 538 bp). To distinguish between *Clec11a*$^{+/-}$and *Clec11a*$^{-/-}$ mice, the following primers were used: 5'-GAG GAA GAG GAA ATC ACC ACA GC-3' and 5'-TTG CAC TGA GTC GCG GGT G-3' (*Clec11a*$^{+/-}$: 482 bp; *Clec11a*$^{-/-}$: no amplification product).

## Statistical analysis

The statistical significance of differences between the two treatments was assessed using two-tailed Student's t tests. The statistical significance of differences among more than two groups was assessed using one-way ANOVAs with Tukey's multiple comparison tests. The statistical significance of differences in long-term competitive reconstitution assays was assessed using two-way ANOVAs with Sidak's multiple comparison tests. All data represent mean ± SD. *p<0.05, **p<0.01, ***p<0.001.

# Acknowledgements

SJM is a Howard Hughes Medical Institute (HHMI) Investigator, the Mary McDermott Cook Chair in Pediatric Genetics, the Kathryn and Gene Bishop Distinguished Chair in Pediatric Research, the director of the Hamon Laboratory for Stem Cells and Cancer and a Cancer Prevention and Research Institute of Texas Scholar. RY was supported by a Damon Runyon Cancer Research Foundation fellowship. We thank Nicolas Loof and the Moody Foundation Flow Cytometry Facility, Kristen Correll for mouse colony management, Christopher Chen for biomechanical analysis of bones, Malea Murphy for assistance with confocal imaging, Ying Liu, Jingya Wang and Jerry Q Feng at the Texas A and M University Baylor College of Dentistry for assistance with microCT experiments. This work was supported by the Cancer Prevention and Research Institute of Texas.

# Additional information

### Competing interests

SJM: Senior editor, *eLife*. The other authors declare that no competing interests exist.

### Funding

| Funder | Author |
| --- | --- |
| Howard Hughes Medical Institute | Sean J Morrison |
| Cancer Prevention and Research Institute of Texas | Sean J Morrison |

The funders had no role in study design, data collection and interpretation, or the decision to submit the work for publication.

### Author contributions

RY, SJM, Conception and design, Acquisition of data, Analysis and interpretation of data, Drafting or revising the article; BS, Acquisition of data, Analysis and interpretation of data

### Author ORCIDs

Sean J Morrison, http://orcid.org/0000-0003-1587-8329

## Ethics

Animal experimentation: This study was performed in strict accordance with the recommendations in the Guide for the Care and Use of Laboratory Animals of the National Institutes of Health. All procedures were approved by the UT Southwestern Medical Center Institutional Animal Care and Use Committee. Animal protocol number: 2016-101334-G

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
