## [Decision Letter]

Thank you for submitting your article "Clec11a is an osteogenic factor that promotes the maintenance of the adult skeleton" for consideration by *eLife*. Your article has been reviewed by three peer reviewers, including Ophir D Klein (Reviewer #1), and the evaluation has been overseen by Janet Rossant as the Senior Editor and Reviewing Editor.

The reviewers have discussed the reviews with one another and the Reviewing Editor has drafted this decision to help you prepare a revised submission.

Summary:

In this elegant and well-executed manuscript, the Morrison lab reports on their exciting discovery that a C-type lectin domain protein, Clec11a, promotes osteogenesis. Clec11a maintains the adult skeleton by promoting osteogenic differentiation of mesenchymal progenitors. The authors first determined that Clec11a is highly expressed in skeletal lineage cells by gene and protein expression analyses. They then employed the CRISPR-Cas9 system to generate a Clec11a knock out mouse to test if Clec11a is necessary for hematopoiesis and osteogenesis in vivo. They found that Clec11a deficiency led to weaker bones under conditions of normal homeostasis and also after fracture healing. The authors also tested whether administration of recombinant Clec11a is sufficient to prevent and reverse osteoporosis using micro-CT and histomorphometric analyses. The identification of Clec11a as a novel agonist of bone maintenance and fracture healing is of high interest and the data presented are very compelling. While the study does not define a clear molecular mechanism of action for Clec11a, multiple models of osteoporosis were used to demonstrate the efficacy of rClec11a, greatly strengthening the clinical relevance of the study.

Major revisions required:

Although there was overall support for the findings, there were concerns from the reviewers about the robustness of the data on the bone phenotypes, specifically about the rigor of the µCT and histomorphometry data. Reviewer 1 noted that there are standard protocols for assessing and reporting mouse bone microstructure (for example: Bouxsein et al., 2010 JBMR. Guidelines for assessment of bone microstructure in rodents using micro-computed tomography). As presented here, the explanations for the exact location of the volume of interest and the method used to define trabecular and cortical bone regions are vague. If indeed the same number of slices were chosen for every timepoint, then it is likely that different anatomical regions within the growing femur were assessed. Details regarding image processing, including the algorithm used for image filtration and the approach used for calibration and image segmentation, should be added. Many of the units used by the authors to describe the various bone features differ from those recommended in standard protocols.

In addition, there was concern as to whether the sample size of mice analyzed was sufficient to support the conclusions in all cases. Typically, unless the effect size is substantial, labs need to study 8 to 10 mice/genotype/sex/treatment group for most bone parameters. Because you present "representative" images in this paper and no clear indication of the number or sex of animals included in each group, the reviewers were not convinced that all the reported in vivo effects represent true positive results. It will be important for you to address these concerns as far as is possible within the two month time frame, or focus your results on those experiments where you have a fully powered cohort.

Other concerns include explaining why mice with similar ages and genotypes have very different bone measures (e.g., BV/TV) in different figures and a need for a better description of the recombinant protein you produced and administered. Downstream experiments (beyond the scope of the present paper) could then determine if the effect on bone cells is direct or indirect, why fracture healing is delayed (osteoblast recruitment or transdifferentiation failure), and whether recombinant CLEC11A can be used as a therapeutic for common skeletal diseases such as fractures and osteoporosis.

Overall there was enthusiasm for the finding of a potentially novel osteogenic factor.

---

## [Author Response]

*[…] Major revisions required:*

*Although there was overall support for the findings, there were concerns from the reviewers about the robustness of the data on the bone phenotypes, specifically about the rigor of the µCT and histomorphometry data. Reviewer 1 noted that there are standard protocols for assessing and reporting mouse bone microstructure (for example: Bouxsein et al., 2010 JBMR. Guidelines for assessment of bone microstructure in rodents using micro-computed tomography). As presented here, the explanations for the exact location of the volume of interest and the method used to define trabecular and cortical bone regions are vague. If indeed the same number of slices were chosen for every timepoint, then it is likely that different anatomical regions within the growing femur were assessed. Details regarding image processing, including the algorithm used for image filtration and the approach used for calibration and image segmentation, should be added.*

We believe we followed the standard protocols described by Bouxsein et al. (see details below). We have updated the text of the manuscript to clarify more precisely the locations of the volume measurements. We have included more details regarding the number of slices that were analyzed. Bone growth was very limited in the adult mice analyzed in our study so we do not believe that any of our results are confounded by changes in anatomical regions caused by bone growth. We have also updated the text to include more details regarding image processing and analysis.

For the femur trabecular bones, we started analyzing slices at the bottom of the distal growth plate, where the epiphyseal cap structure completely disappeared, and continued for 100 slices (3.5 μm/slice) toward the proximal end of the femur. Contours were drawn manually a few voxels away from the endocortical surface to define trabecular bones in the metaphysis, as suggested by the guideline (Figure 4; JBMR 25:1468). For the femur cortical bones, 100 slices (3.5 μm/slice) in the mid-diaphysis were analyzed. The same anatomical regions were analyzed for all the samples throughout the manuscript. We have added details of microCT measurements and analyses as requested.

*Many of the units used by the authors to describe the various bone features differ from those recommended in standard protocols.*

In the original manuscript we used ratios instead of “%” for Bone volume/Total volume and Cortical area/Total area (i.e. 0.05 rather than 5%). We used “per mm” instead of “1/mm” (as recommended by Bouxsein et al.) for trabecular number and a few other measurements. We have now changed the units to be identical to those recommended by Bousein et al., though these changes have no effect on the data or the conclusions.

*In addition, there was concern as to whether the sample size of mice analyzed was sufficient to support the conclusions in all cases. Typically, unless the effect size is substantial, labs need to study 8 to 10 mice/genotype/sex/treatment group for most bone parameters. Because you present "representative" images in this paper and no clear indication of the number or sex of animals included in each group, the reviewers were not convinced that all the reported* in vivo *effects represent true positive results. It will be important for you to address these concerns as far as is possible within the two month time frame, or focus your results on those experiments where you have a fully powered cohort.*

There are multiple issues that bear clarification here:

1) In the original manuscript, we specified the number of mice per genotype and the number of independent experiments for each figure panel in the figure legends.

2) Although we did not have n=8 mice/sex/genotype in all experiments, all of our conclusions were based on statistically significant differences in which the statistical tests took into account variability within treatments as well as numbers of replicates. Many of the effects reported in our manuscript were large effects. For example, Clec11a deficiency has a large negative effect on bone volume in vivo (e.g. Figure 2) and bone formation in culture (e.g. Figure 3). Treatment with recombinant Clec11a has large positive effects on bone formation in culture (e.g. Figure 5) and bone formation in vivo (e.g. Figure 5). The fact that we consistently observed statistically significant differences in many different kinds of experiments, with both gain and loss of function assays that showed opposite results, in vitro and in vivo, makes it exceedingly unlikely that the effects could have been observed by chance.

3) The representative images in the figures were usually accompanied by bar graphs that showed all of the data from all of the experiments (with numbers of replicates and experiments in the figure legend) as well as statistical analysis.

Most of our experiments require more than 2 months to perform. However, we have performed new experiments and added more replicates for in vivo administration of rClec11a in wild-type mice (new Figure 5), and for prevention of ovariectomy-induced osteoporosis by rClec11a (new Figure 6). The new data were consistent with the old data: in vivo administration of rClec11a promotes bone formation in wild-type mice and in ovariectomized mice.

*Other concerns include explaining why mice with similar ages and genotypes have very different bone measures (e.g., BV/TV) in different figures and a need for a better description of the recombinant protein you produced and administered.*

Thank you for calling our attention to this. We have revised our presentation of these data to avoid confusion. We used both males and females of each genotype in Figure 2, whereas only females were used in Figure 5, Figure 6 and Figure 7 (e.g. ovariectomy studies could only be performed on females). Since adult male mice have significantly higher BV/TV values than female mice (see Figure 3; JBMR 22:1197), it is not surprising that the 2 month-old BV/TV value in the control mice of Figure 2 (around 10%) is higher than in the rest of the manuscript (4-8%). Thus, the differences in BV/TV values in different figures mainly reflect expected biological differences, not experimental error. We have revised the manuscript to make clear the sexes that were used in each experiment. We have also added more detailed information on the recombinant protein and how it was purified.